# Lateral Ridge Augmentation with Guided Bone Regeneration Using Particulate Bone Substitutes and Injectable Platelet-Rich Fibrin in a Digital Workflow: 6 Month Results of a Prospective Cohort Study Based on Cone-Beam Computed Tomography Data

**DOI:** 10.3390/ma14216430

**Published:** 2021-10-27

**Authors:** Maoxia Wang, Xiaoqing Zhang, Yazhen Li, Anchun Mo

**Affiliations:** 1State Key Laboratory of Oral Diseases, National Clinical Research Center for Oral Diseases, Department of Oral Implantology, West China Hospital of Stomatology, Sichuan University, Chengdu 610041, China; wangmaoxia@stu.scu.edu.cn (M.W.); 2017224035151@stu.scu.edu.cn (X.Z.); 2State Key Laboratory of Oral Diseases, National Clinical Research Center for Oral Diseases, Department of Orthodontics, West China Hospital of Stomatology, Sichuan University, Chengdu 610041, China; 2019324035095@stu.scu.edu.cn

**Keywords:** computer-aided surgery, guided bone regeneration, lateral ridge augmentation, injectable platelet-rich fibrin

## Abstract

This study aimed to test whether or not a digital workflow for GBR with particulate bone substitutes and injectable platelet-rich fibrin improved the thickness of the hard tissue compared to the conventional workflow. 26 patients in need of lateral bone augmentation were enrolled. GBR with particulate bone substitutes and injectable platelet-rich fibrin was performed in all patients. Patients were divided into two groups: control (conventional workflow; n = 14) and test (digital workflow; n = 12). CBCT scans were performed before surgery, immediately after wound closure, and 6 months post-surgery, and the labial thickness of the hard tissue (LT) was assessed at 0–5 mm apical to the implant shoulder (LT_0_–LT_5_) at each time point. A total of 26 patients were included in this study. After wound closure, the test group showed significantly greater thickness in LT_0_–LT_2_ than the control group (LT_0_: test: 4.31 ± 0.73 mm, control: 2.99 ± 1.02 mm; LT_1_: test: 4.55 ± 0.69 mm, control: 3.60 ± 0.96 mm; LT_2_: test: 4.76 ± 0.54 mm, control: 4.05 ± 1.01 mm; *p* < 0.05). At 6 months, significant differences in LT_0_–LT_1_ were detected between the groups (LT_0_: test: 1.88 ± 0.57 mm, control: 1.08 ± 0.60 mm; LT_1_: test: 2.36 ± 0.66 mm, control: 1.69 ± 0.58 mm; *p* < 0.05). Within the limitations of this study, the use of digital workflow in GBR with particulate bone substitutes and i-PRF exerted a positive effect on the labial thickness of hard tissue in the coronal portion of the implant after wound closure and at 6 months.

## 1. Introduction

Alveolar bone remodeling and resorption processes following tooth removal induce gradual changes in both soft and hard tissue dimensions [1]. Prosthetically driven bone regeneration procedures are therefore required to reconstruct ridge contours for implant placement in the prosthetically ideal position and achieve optimized esthetic and functional outcomes [2]. Numerous techniques have been used to increase the residual bone volume, such as guided bone regeneration (GBR), autogenous bone block grafting, ridge splitting, and distraction osteogenesis [3,4,5,6]. GBR has become a routine and reliable therapy for the treatment of horizontal bone defects [7,8,9].

Resorbable collagen membranes in combination with particulate bone substitutes are widely used in GBR procedures with long-term clinical success, as they have the advantage of good tissue integration and low complication rates [10]. However, the major drawback of particulate bone substitutes and collagen membranes is their poor mechanical properties [11]. Compressive forces generated during the wound closure and healing period can lead to the collapse of collagen membranes and the apical displacement of particulate bone substitutes, thereby impairing space maintenance and new bone formation [12,13]. Moreover, the poor mechanical properties of bone grafts lead to difficulties in achieving appropriate bone graft contours precisely during the implementation of surgery, and the results of bone regeneration rely largely on operators’ experience.

A variety of strategies have been proposed to improve the volume stability of bone grafts. Strategies aiming at enhancing the mechanical properties of grafting materials have been proven to be effective [12,14,15]. Mertens et al. reported that bone blocks can provide greater graft stability than particulate bone substitutes in the treatment of one-wall horizontal bone defects [16]. The use of soft-block xenografts instead of particulate bone grafts also has been shown to reduce the apical displacement of bone grafts [17]. Meanwhile, injectable platelet-rich fibrin (i-PRF) has been recommended to agglutinate particulate bone substitutes in GBR procedures [18]. The mixture of i-PRF and particulate bone substitutes (i-PRF block) can become another type of soft-block bone graft with increased compressive resistance and operability, providing a sustained release of multiple growth factors facilitating wound healing and new bone formation [19,20]. However, as most published reports about i-PRF blocks are case reports, the clinical outcome of i-PRF blocks in GBR is required for further research.

Additionally, the sufficient volume stability of bone grafts offers the possibility of controlling the shape of bone grafts precisely through digital techniques. Several digital procedures, including individualized titanium mesh and customized bone blocks, have been used in bone augmentation and rendered good clinical outcomes. In our previous study, we introduced a novel GBR procedure using digital surgical templates for shaping the i-PRF block based on the concept of prosthetic-guided regeneration, and it was confirmed that this procedure contributes to achieving an appropriate bone graft contour after wound closure [21]. However, there is still a paucity of evidence for the clinical efficacy of this GBR procedure, as our previous study only focused on the contour of bone grafts after wound closure.

Therefore, this prospective cohort study aimed to test whether digital workflow provides better radiological outcomes compared to the conventional workflow in GBR with particulate bone substitutes and i-PRF. The null hypothesis was that there is no difference between these workflows.

## 2. Materials and Methods

### 2.1. Study Population

The STROBE guidelines for reporting observational cohort studies were followed in the present study. This study was designed as a prospective cohort study and was conducted at the Department of implant dentistry, West China Hospital of Stomatology, Sichuan University, China from April 2019 to December 2020. The study was conducted following the guidelines of the Declaration of Helsinki, and approved by the ethical committee of West China Hospital of Stomatology, Sichuan University (WCHSIRB-D-2019-068). Informed consent was obtained from all patients involved in the study.

#### 2.1.1. Inclusion Criteria

Male or female patients aged 18 to 60 years (including 18 and 60 years).Presence of a three-wall or two-wall horizontal bone defect in the anterior region.Bone augmentation was applied >3 months after tooth extraction.Good general health.Patients were willing to participate in this study and signed the informed consent form.

#### 2.1.2. Exclusion Criteria

Uncontrolled systemic diseases.Presence of acute infection.Uncontrolled periodontal disease.Heavy smokers (>20 cigarettes per day).Females in pregnancy or lactation.

### 2.2. Surgical Procedures

An intraoral scan was performed before surgery for preoperative design (3Shape TRIOS^®^, 3Shape, Copenhagen K, Denmark) and a diagnostic wax-up was generated on the intraoral scan. Then the standard tessellation language (STL) files of the intraoral scan and diagnostic wax-up were overlapped above the DICOM files of the preoperative CBCT images in Simplant software (Simplant Pro 17.01, Dentsply Sirona, York, PA, USA). Virtual implants of 3.5 mm diameter and 10 mm length were placed under the guidance of the ideal future implant restoration [22,23]. Data of preoperative design were saved as an SPR file, which was named “baseline.spr”.

A copy of “baseline.spr” was created, and the 3.5 mm diameter and 10 mm-length virtual implants were replaced with specific virtual implants (Bone Level Titanium SLA, Institut Straumann AG, Basel, Switzerland; NobelActvie, Nobel Biocare, Göteborg, Sweden) of proper size. The patients were free to choose implant systems according to their own will. The direction, labial position, and depth of the specific virtual implants were the same as the 3.5 mm diameter and 10 mm-length virtual implants (Figure 1). A tooth-supported surgical guide plate was fabricated based on the copy of “baseline.spr”.

All surgical procedures were performed by an experienced surgeon (MA) under local anesthesia. A mid-crestal incision was made in the gingiva with one or two vertical releasing incisions and the mucoperiosteal flap was elevated. A periosteal releasing incision was performed to achieve a tension-free primary closure. After cortical perforation, the implant sites were prepared according to the manufacturer’s recommendations. Two-piece implants of proper size were inserted in a prosthetically ideal position under the guidance of surgical templates. Then, a cover screw was connected to the implant. A staged approach of implant placement was performed at the sites without enough primary stability (<15 N.cm). As the study was designed as an observational study, the patients were divided into two groups only according to their willingness to use a digital surgical template for bone augmentation:

Control group: After implant placement, two tubes of 10 mL of venous blood were collected from the patients and centrifuged (700 rpm for 3 min on a Trausim AiPRF-08 Centrifuge, RCF-max = 60 g) [24]. The upper yellow liquid layer (i-PRF) was collected with a sterile syringe. Then the i-PRF was mixed with particulate bone substitutes (Bio-Oss, Geistlich Pharma AG, Wolhusen, Switzerland; Bio-Gene, Beijing Datsing Bio-tech Co., Beijing, China) to form i-PRF block [19]. Then the i-PRF block was placed on the defect and shaped freehand. A collagen membrane (Bio-Gide, Geistlich Pharma AG, Wolhusen, Switzerland) was placed over the i-PRF block and fixed with several 4 mm titanium pins (MatrixMIDFACE, Depuy Synthes, Warsaw, NY, USA) (Figure 2).

Test group: the surgical procedure was similar to the control group. However, a digital simulation of bone graft contour aiming at reconstructing the ideal alveolar contour and ensuring enough bone for implant placement, was performed based on the pre-operative CBCT data before surgery (Mimics 20.0, Materialise, Leuven, Belgium). The virtual bone graft contour was over-thickened to achieve 1.5 mm of over-augmentation, after which a two-piece tooth-supported surgical template was designed by Mimics software and manufactured by 3D printing technology (ProJet MJP 2500Plus, 3D Systems, Inc., Rock Hill, SC, USA). During the surgery, the i-PRF block was placed into the defect under the guidance of the surgical template to form a customized i-PRF block. A collagen membrane was covered over the customized i-PRF block and fixed with several 4 mm titanium pins (Figure 3).

Finally, the flaps were sutured with horizontal mattress sutures and single interrupted sutures. The sutures were removed 2 weeks post-surgery. Patients were recalled for follow-up visits 2 weeks, 3 months, and 6 months post-surgery. All the complications such as wound dehiscence and infection were recorded.

### 2.3. Radiographic Evaluation

All the patients received CBCT scanning before surgery (T0), immediately after wound closure (T1), and 6 months after surgery (T2) under the same projection conditions (3DAccuitomo 170, J. Morita Mfg. Corp., Kyoto, Japan). The images were acquired with the following protocol: acceleration voltage, 90 kV; beam currency, 5 mA; acquisition time: 17.5 s; FOV diameter, 140 mm; FOV height, 100 mm; and voxel size, 0.25 mm.

Based on the concept of prosthetic-guided regeneration, the predesigned virtual implants in the “baseline.spr” were used as a reference for radiographic evaluation. The DICOM files of the postoperative CBCT scans at T1 and T2 were converted to STL files, respectively, using Simplant software and superimposed on the preoperative CBCT in the “baseline.spr”, while anatomical structures (such as anterior nasal spine and posterior nasal spine) were used for superimposition. The superimposition of CBCT at T0–T2 was following the method proposed by Jiang et al. [25].

Measurements were performed in the bucco-oral cross-sectional image perpendicular to the virtual implant. At each time point (T0, T1, and T2), the distance between the implant and the labial outline of the hard tissue, which represented the labial thickness of the hard tissue, was measured at the implant shoulder (LT_0_) and 1, 2, 3, 4 and 5 mm (LT1–LT_5_) apical to the implant shoulder (Figure 4). Graft gain was defined as the difference between LT at T1 and LT at T0. Bone gain was defined as the difference between LT at T2 and LT at T0. Bone resorption was defined as the difference between LT at T1 and LT at T2.

An experienced, calibrated, and blinded investigator (LY) designed the position of implants and performed all measurements. All the parameters were measured twice and averaged. In cases of multiple sites per patient, only one site was randomly selected by research staff (LY) for measurement. Ten randomly selected sites were remeasured to determine intra-observer reliability after a one-month interval. The intra-observer reliability was tested by the intra-class correlation coefficient (ICC) for the parameters that resulted in good agreement (ICC ranged from 0.957 to 0.995).

### 2.4. Sample Size Calculation

PASS software (PASS 15, NCSS, LLC. Kaysville, UT, USA,) was used for sample size calculation. In this study, the labial thickness of the hard tissue at the implant shoulder (LT_0_) was considered the primary outcome. Considering 1 mm as a clinically relevant difference between groups [26] and a 20% drop-out rate, a sample size of 24 patients (12 patients per group) was required to achieve 80% power to reject the null hypothesis of equal means (standard deviation of 0.80 mm according to the study of Benic [14]) with a significance level of 0.05 in a design with 3 repeated measurements having a compound symmetry covariance structure, while the correlation between observations on the same subject was estimated to be 0.700.

### 2.5. Statistical Analysis

Statistical analyses were performed using SPSS 20.0 software (IBM Company, Armonk, New York, NY, USA). Data were summarized by descriptive statistics. Quantitative data were reported as means ± SD. A Shapiro–Wilk test was applied to test normality. Longitudinal data were analyzed with a generalized linear mixed effect model (GLMM). LT0–LT5 were considered as the dependent variables, respectively. Groups, time (T0-T2), and groups ∗ time interaction were considered as fixed factors. Patient effects were modeled as random effects. The diagonal type covariance structure was determined to provide the best covariance model fit based on the Akaike and Bayesian information criteria. Planned contrasts were used to compare the two groups at each time point. The *p* values were adjusted for multiple comparisons using the Bonferroni method.

For the statistical comparisons of graft gain, bone gain, and bone resorption, a two-sample t-test was used for data with a normal distribution, and a Mann–Whitney U-test was used for data with a non-normal distribution.

The impact of all variables on the primary outcome (LT0) was analyzed using GLMM. LT0 was considered as the dependent variable. Time, groups ∗ time interaction, gender ∗ time interaction, age ∗ time interaction, location ∗ time interaction, jaw ∗ time interaction, defect type ∗ time interaction, implant type ∗ time interaction, and bone substitutes ∗ time interaction were considered as fixed factors. Patient effects were modeled as random effects. The statistical test level was set as 0.05.

## 3. Results

### 3.1. Patients

Twenty-six patients were included in this study (12 in the test group and 14 in the control group). No patients dropped out during the follow-up period. Patient demographics and the characteristics of the augmented sites for each group are reported in Table 1. No statistically significant differences existed between the two groups.

### 3.2. Soft Tissue Condition

During 6 months of follow-up, soft tissue healing was uneventful in all patients.

### 3.3. Radiographic Outcomes

The results of the labial thickness of the hard tissue at each time point are visualized in Table 2. There were no statistically significant differences in LT at baseline between the groups (*p* > 0.05). After wound closure, the median LT0 amounted to 4.31 mm (mean ± SD: 4.31 ± 0.73 mm) in the test group and 2.99 mm (mean ± SD: 2.99 ± 1.02 mm) in the control group. The differences in LT0–LT2 between the control and test groups were statistically significant (*p* < 0.05). At 6 months, the test group showed significantly greater thickness in LT0 (mean ± SD: 1.88 ± 0.57 mm) and LT1 (mean ± SD: 2.36 ± 0.66 mm) when compared with the control group (*p* < 0.05).

The results of graft gain, bone gain, and bone resorption and the differences between the groups are displayed in Table 3. The test group achieved significantly more graft gain and bone gain than the control group at level LT0–LT3 (*p* < 0.05). In terms of bone resorption, no statistical difference was detected between the groups at all levels (*p* > 0.05).

All data from the two groups were combined to explore the impact of all variables on the primary outcome (LT0) through GLMM. GLMM revealed that gender, age, location, jaw, defect type, implant type, and bone substitutes did not affect the primary outcome (*p* > 0.05), while the time and groups ∗ time interaction had a significant effect on the primary outcome (*p* < 0.05) (Table 4).

## 4. Discussion

The results of the present study demonstrated that the use of digital workflow in GBR exerts a positive effect on the labial thickness of hard tissue in the coronal portion of the implant after wound closure and at 6 months post-surgery.

The volume of bone grafts after wound closure is known to be associated with the bone volume after 6 months of healing in lateral ridge augmentation, as the bone grafts maintain the initial space for new bone formation [27]. Obtaining adequate bone grafts at all levels immediately after surgery is therefore important in lateral bone augmentation. Many studies had been carried out focusing on this point, clinical operation and the volume stability of grafting materials were considered important factors influencing the contour of bone grafts after wound closure [16,17,21].

Our previous study had shown that the instability of freehand operation was unfavorable for achieving adequate labial graft thickness in the coronal portion of implants after wound closure [21]. Targeting the deficiency of freehand operation, the digital workflow was put forward to control the contour of bone grafts during the surgery. Based on the concept of prosthetic-guided regeneration, a digital simulation of the bone graft contour was conducted to determine the position of future prostheses and implants, after which a two-piece surgical template was designed and fabricated. Thus, the grafting materials could be properly shaped under the guidance of the surgical template during the surgery. In our previous study, the digital workflow had been proven to contribute to achieving greater labial graft thickness in the coronal portion of bone grafts immediately after surgery. Similar results were obtained in the present study: the test group showed significantly greater LT at level LT_0_–LT_2_ after wound closure. The LT_0_ after wound closure was 4.31 ± 0.73 mm in the test group, while the LT_0_ after wound closure was 2.99 ± 1.02 mm in the control group. Although the surgeon was instructed to achieve 1 mm of over-augmentation during the freehand surgery, it was hard to ensure sufficient graft thickness at all levels. The control group showed a larger standard deviation in LT at all levels after wound closure, which indicated the instability of freehand operation.

Furthermore, the short-term outcomes of the digital workflow were exhibited in the present study. After 6 months of healing, the test group showed significantly greater LT at levels LT_0_ and LT_1_ when compared to the control group (LT_0_: test: 1.88 ± 0.57 mm, control: 1.08 ± 0.60 mm; LT_1_: test: 2.36 ± 0.66 mm, control: 1.69 ± 0.58 mm). In terms of graft gain and bone gain, the test groups achieved significantly more graft gain and bone gain at levels LT_0_–LT_3_ (*p* < 0.05). As the bone grafts were the same for both groups, the two groups exhibited similar bone resorption during the 6-months healing period (*p* > 0.05). Thus, the better performance and bone gain of the test group in LT at 6 months could be due to the advantages of the digital workflow in the control of the bone graft contour.

The volume stability of grafting materials was another important factor influencing the clinical outcomes of GBR. Although digital techniques can be used to manage the contour of bone grafts, sufficient mechanical properties of grafting materials were required to maintain the space. Block grafts have been proven to be a good choice to withstand the compressive forces generated during wound closure [16]. However, the precise trim of bone blocks and soft tissue management was challenging in clinical practice, where bone blocks were associated with a high risk of mucosal dehiscence [28,29]. Thus, collagenated bovine bone mineral—a kind of soft-block grafting material—was recommended for GBR and renders good volume stability [17]. Similar to collagenated bovine bone minerals, the mixture of i-PRF and particulate bone substitutes has been utilized in bone augmentation [19,30]. As our results show, graft granules were entrapped by the fibrin matrix from i-PRF once the i-PRF block was formed. Then the mechanical properties of particulate bone substitutes were improved. A recent in vitro study compared the mechanical properties of different composite bone grafts and showed that the compressive resistance of the i-PRF block was 8.75-fold higher than the mixture of physiological water and graft granules [31]. Additionally, a i-PRF block can be shaped easily and therefore is an excellent medium for the digital workflow. Another rational reason for the use of i-PRF in bone augmentation lies in its biological properties. The liquid i-PRF can be transformed as a three-dimensional fibrin scaffold with leukocytes, platelets, type I collagen, osteocalcin, and plentiful growth factors [24,32]. Multiple autologous factors (such as vascular endothelial growth factor, platelet-derived growth factor, and transforming growth factor-beta) can be sustainedly released to promote the temporal-spatial vascular formation required for new bone formation [33]. Moreover, i-PRF has been proven to stimulate osteogenesis by influencing the migration, proliferation, and differentiation of human osteoblasts [20]. One recent study reported that PRF showed promising anti-inflammatory activity and could shift macrophage polarization toward an M2 phenotype, while the activation of M2 macrophages around biomaterials can promote bone regeneration [34,35].

Concerning the research methods, some limitations have to be addressed. First, many confounders (such as location, jaw, defect type, implant type, and bone substitutes) were involved in this study. However, no significant difference was detected in the baseline characteristics between the test group and the control group, which guarantees good comparability between the two groups. GLMM was performed to explore the impact of these factors on the primary outcome. Results showed that gender, age, location, jaw, defect type, implant type, and bone substitutes did not cause confounding effects on the primary outcome. Moreover, outcome assessor blinding was implemented to minimize the risk of detection bias, and no ICC value for the measurements was less than 0.95 in the current study, which indicates excellent reliability and repeatability. Finally, this study had a short follow-up period. Further studies with a reasonable and scientific design were needed to confirm the results in the present study.

## 5. Conclusions

Within the limitations of the study, our results have shown that the use of digital workflow in GBR with particulate bone substitutes and i-PRF exerts a positive effect on the labial thickness of hard tissue in the coronal portion of the implant after wound closure and at 6 months post-surgery.

## Figures and Tables

**Figure 1 materials-14-06430-f001:**
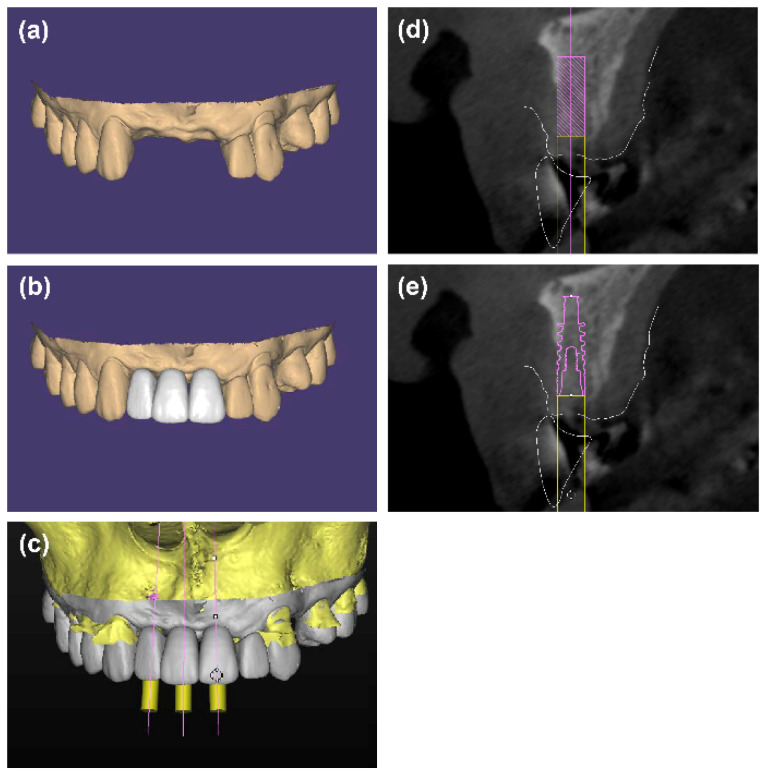
(**a**–**d**) 3.5 mm diameter and 10 mm-length virtual implants were placed in a prosthetically ideal position under the guidance of the diagnostic wax-up. (**e**) The 3.5 mm diameter and 10 mm-length virtual implants were replaced with specific virtual implants of the proper size to fabricate the surgical guide plate for implant placement.

**Figure 2 materials-14-06430-f002:**
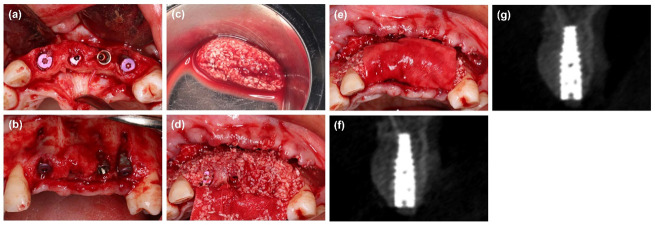
(**a**,**b**) The labial defect could be observed. (**c**) Particulate bone substitutes were mixed with i-PRF to form i-PRF blocks. (**d**,**e**) The i-PRF block was placed into the bone defect and covered with a collagen membrane, and titanium pins were used to fixate the membrane. (**f**) Radiographic cone-beam CT view immediately after wound closure. (**g**) Radiographic cone-beam CT view at 6 months.

**Figure 3 materials-14-06430-f003:**
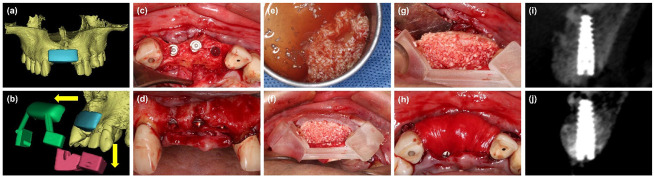
(**a**) Digital simulation of the bone graft contour before surgery. (**b**) A two-piece surgical template, which consists of two parts: the coronal part (red) for retention and the labial part (green) for shaping the bone grafts, was fabricated based on the digital model. the template can be removed without disrupting the graft material. (**c**,**d**) The labial defect could be observed. (**e**) Particulate bone substitutes were mixed with i-PRF. (**f**,**g**) The mixture of i-PRF and particulate bone substitutes was placed into the defect under the guidance of the surgical template. (**h**) The customized i-PRF block was covered with a collagen membrane and fixed with pins. (**i**) Radiographic cone-beam CT view immediately after wound closure. (**j**) Radiographic cone-beam CT view at 6 months.

**Figure 4 materials-14-06430-f004:**
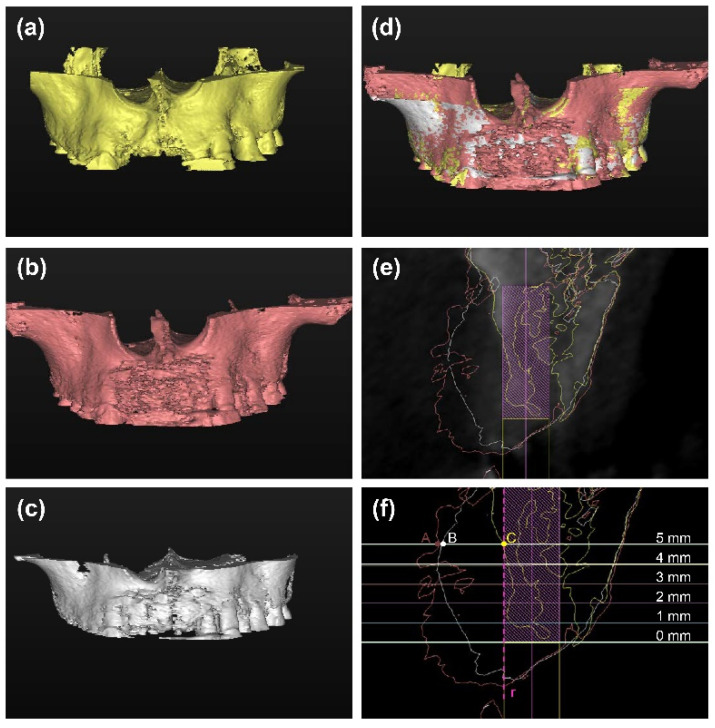
(**a**–**d**) The DICOM files of the postoperative CBCT scans at T1 ((**b**), red) and T2 ((**c**), white) were converted to STL files, respectively, and superimposed on the preoperative CBCT ((**a**), yellow). (**e**) The bucco-oral cross-sectional image perpendicular to the virtual implant was used for measurements. Yellow line: the outline of the hard tissue before surgery (T0); red line: the outline of the hard tissue after wound closure (T1); white line: the outline of the hard tissue at 6 months (T2). (**f**) The labial thickness of the hard tissue (LT) was measured at the implant shoulder (LT_0_) and 1, 2, 3, 4 and 5 mm (LT_1_–LT_5_) apical to the implant shoulder at each time point. The labial outline of the virtual implant was drawn as a reference line “r”. Six lines perpendicular to line “r” at different levels were labially intersected with the 3 outlines (eg, at level LT5, the distance from point “A”, “B”, “C” to line “r” was LT at T1, T2, T3, respectively). The LT was labeled as “0” when the outline of the hard tissue was in the palatal side of the line “r”.

**Table 1 materials-14-06430-t001:** Descriptive statistics for each group.

	Test	Control	*p*-Value
Patient demographics			
Male/Female	6/6	7/7	1.000
Smoking/No smoking	2/10	1/13	0.580
Mean age ± SD	38.92 ± 14.12	41.71 ± 13.24	0.607
Information about augmented sites			
Location CI/LI	6/6	6/8	1.000
Jaw Maxilla/Mandible	11/1	12/2	1.000
Three-wall defect/Two-wall defect	8/4	11/3	0.665
Implant			0.583
NobelActive	5	8	
Bone Level Titanium SLA	2	3	
Staged implant placement	5	3	
Bone substitutes			0.713
Bio-Oss (xenograft)	5	7	
Bio-Gene (allograft)	7	7	

Abbreviations: SD, standard deviation; CI, central incisor; LI, lateral incisor.

**Table 2 materials-14-06430-t002:** Results of the labial thickness of the hard tissue at different vertical levels (mm).

Level	Before Surgery (T_0_)			Immediately after Surgery (T_1_)			6 Months after Surgery(T_2_)		
	Test	Control	*p*-Value	Test	Control	*p*-Value	Test	Control	*p*-Value
	Mean ± SD								
LT_0_	0.03 ± 0.11	0.14 ± 0.36	0.316	4.31 ± 0.73	2.99 ± 1.02	0.000 *	1.88 ± 0.57	1.08 ± 0.60	0.001 *
LT_1_	0.15 ± 0.51	0.38 ± 0.58	0.256	4.55 ± 0.69	3.60 ± 0.96	0.008 *	2.36 ± 0.66	1.69 ± 0.58	0.009 *
LT_2_	0.20 ± 0.48	0.40 ± 0.55	0.320	4.76 ± 0.54	4.05 ± 1.01	0.034 *	2.62 ± 0.81	2.10 ± 0.55	0.069
LT_3_	0.09 ± 0.27	0.33 ± 0.51	0.144	5.01 ± 0.65	4.42 ± 1.06	0.072	2.97 ± 0.77	2.49 ± 0.73	0.150
LT_4_	0.17 ± 0.32	0.40 ± 0.60	0.254	5.03 ± 0.67	4.67 ± 1.07	0.288	3.12 ± 0.88	2.65 ± 0.78	0.143
LT_5_	0.16 ± 0.34	0.47 ± 0.68	0.160	5.13 ± 0.63	4.94 ± 1.15	0.594	3.26 ± 0.86	2.67 ± 0.81	0.090

Abbreviations: SD, standard deviation; LT_X_, the labial thickness of the hard tissue measured x mm apical to the implant shoulder; * statistically significant.

**Table 3 materials-14-06430-t003:** Results of graft gain, bone gain, and bone resorption at the labial side of the virtual implant (mm).

Level	Graft Gain (T_1_–T_0_)			Bone Gain (T_2_–T_0_)			Bone Resorption(T_1_–T_2_)		
	Test	Control	*p*-Value	Test	Control	*p*-Value	Test	Control	*p*-Value
	Mean ± SD								
LT_0_	4.28 ± 0.75	2.85 ± 0.96	0.000 *	1.85 ± 0.56	0.94 ± 0.53	0.000 *	2.43 ± 0.98	1.91 ± 1.21	0.243
LT_1_	4.41 ± 0.85	3.22 ± 0.89	0.002 *	2.21 ± 0.60	1.31 ± 0.67	0.002 *	2.19 ± 0.95	1.91 ± 1.15	0.506
LT_2_	4.55 ± 0.61	3.65 ± 0.90	0.007 *	2.42 ± 0.86	1.70 ± 0.53	0.017 *	2.13 ± 0.94	1.95 ± 1.14	0.742
LT_3_	4.92 ± 0.69	4.09 ± 0.78	0.009 *	2.87 ± 0.90	2.15 ± 0.84	0.009 *	2.04 ± 0.96	1.93 ± 1.15	1.000
LT_4_	4.85 ± 0.60	4.27 ± 0.90	0.070	2.95 ± 1.15	2.25 ± 0.98	0.085	1.90 ± 1.10	2.02 ± 1.10	0.462
LT_5_	4.97 ± 0.53	4.47 ± 1.05	0.150	3.10 ± 1.09	2.20 ± 1.07	0.045	1.87 ± 1.04	2.27 ± 1.24	0.274

Abbreviations: SD, standard deviation; LT_X_, the labial thickness of the hard tissue measured x mm apical to the implant shoulder; * statistically significant.

**Table 4 materials-14-06430-t004:** Results of generalized linear mixed effect model regarding the primary outcome (labial thickness of the hard tissue at the implant shoulder).

Factors	*p*-Value
Time	0.000
Group ∗ Time	0.000 *
Gender ∗ Time	0.208
Age ∗ Time	0.991
Location ∗ Time	0.435
Jaw ∗ Time	0.210
Defect type ∗ Time	0.089
Implant type ∗ Time	0.806
Bone substitutes ∗ Time	0.067

* Statistically significant.

## Data Availability

The data presented in this study are available on request from the corresponding author. The data are not publicly available due to protecting participant confidentiality.

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
