# Peer review of "Lateral Ridge Augmentation with Guided Bone Regeneration Using Particulate Bone Substitutes and Injectable Platelet-Rich Fibrin in a Digital Workflow: 6 Month Results of a Prospective Cohort Study Based on Cone-Beam Computed Tomography Data"

_materials, 2021, doi:10.3390/ma14216430_

Round 1

Reviewer 1 Report

Congratulations¡¡¡

This research is under the scope of this journal; the topic is relevant for readers, and this research deals with potentially significant knowledge to the field.

I recommend accepting paper with minor revision:

  • There are many mistakes in the references section and in the text
  • The discussion is also misleading. What is the novelty of this paper???
  • Limitations?
  • Conclusions were not totally supported by the data showed.
  • Table legends: Bad descriptions

Author Response

Response to Reviewer 1 Comments

Point 1:

This research is under the scope of this journal; the topic is relevant for readers, and this research deals with potentially significant knowledge to the field.

I recommend accepting paper with minor revision:

There are many mistakes in the references section and in the text

The discussion is also misleading. What is the novelty of this paper???

Limitations?

Conclusions were not totally supported by the data showed.

Table legends: Bad descriptions 

Response 1: We are grateful for the encouraging comments and appreciate the reviewer for the support. And we have revised the manuscript following the reviewer’s advice.

Reviewer 2 Report

The paper presents characteristics of originality and is interesting both from a clinical and technical point of view. The introduction clearly clarifies the problems that will be treated and the possibility of using a sterolithographic template for bone grafting interventions seems interesting to me. The following sections are well conducted. I found only one flaw: when clinical cases are described, the authors refer to bone defects that are evident on clinical and radiographic images. I suggest adding a photograph from the buccal side both after elevation of the flap and with the implants inserted before grafting. I would also change the CBCT images that are cut off, not allowing to evaluate the defect (Fig.!) And CBCT images are even missing in fig. 2

I attach the plagiarism report.

Author Response

Response to Reviewer 2 Comments

Point 1: I found only one flaw: when clinical cases are described, the authors refer to bone defects that are evident on clinical and radiographic images. I suggest adding a photograph from the buccal side both after elevation of the flap and with the implants inserted before grafting.

Response 1: We thank the reviewer’s suggestion. And we now add the images to our revised manuscript (figure 2 and figure 3).

Reviewer 3 Report

The authors conducted interesting research on whether a digital workflow for GBR with particulate bone substitutes and injectable platelet-rich fibrin improved the thickness of the hard tissue compared to the conventional workflow or not. Although this study is well designed and well written, there are some minor modifications that can help to improve it. 

Introduction

Digital workflow should be explained in this section. Current literature regarding digital workflow should be discussed. 

Materials and Methods

The rationale for performing histological evaluation is not related to the aim of the study. In the results also it seems, you histologically assessed the samples only in one group. It needs more clarification. 

Results

1. In materials and methods, you mentioned only heavy smokers were excluded. Thus, it is necessary to report the smoking situation of the patients in both groups since it can be one of the major confounding factors. 

2. Instead of tables 1 and 2, I suggest using bar graphs to present the results which are more reader-friendly and easier to understand.

3. Figure c does not have a scale bar. 

Author Response

Response to Reviewer 3 Comments

Point 1: Digital workflow should be explained in this section. Current literature regarding digital workflow should be discussed. 

Response 1: We thank the Reviewer’s suggestion. And we now add this part to our revised manuscript (lines 65-67).

Point 2: The rationale for performing histological evaluation is not related to the aim of the study. In the results also it seems, you histologically assessed the samples only in one group. It needs more clarification.

Figure c does not have a scale bar.

Response 2: We thank the Reviewer’s suggestion, and we have deleted this part in the revised manuscript.

Point 3: In materials and methods, you mentioned only heavy smokers were excluded. Thus, it is necessary to report the smoking situation of the patients in both groups since it can be one of the major confounding factors.

Response 3: We thank the Reviewer’s suggestion. We now add the smoking situation of the patients in both groups in Table 1.

Point 4: Instead of tables 1 and 2, I suggest using bar graphs to present the results which are more reader-friendly and easier to understand.

Response 4: We thank the Reviewer’s suggestion. However, we prefer to use tables, which can provide more information.

This manuscript is a resubmission of an earlier submission. The following is a list of the peer review reports and author responses from that submission.